# Development and Content Validation of the CEECCA Questionnaire to Assess Ability to Communicate among Individuals with Aphasia Based on the NANDA-I and NOC

**DOI:** 10.3390/healthcare9111459

**Published:** 2021-10-28

**Authors:** Willian-Jesús Martín-Dorta, Pedro-Ruymán Brito-Brito, Alfonso-Miguel García-Hernández

**Affiliations:** 1Primary Care Management of Tenerife, The Canary Islands Health Service, 38003 Santa Cruz de Tenerife, Spain; 2Training and Research in Care, Primary Care Management of Tenerife, The Canary Islands Health Service, 38204 Santa Cruz de Tenerife, Spain; ruymanbrito@gmail.com; 3Department of Nursing, University of La Laguna, 38200 Santa Cruz de Tenerife, Spain; 4Faculty of Healthcare Sciences, Department of Nursing, University of La Laguna, 38200 Santa Cruz de Tenerife, Spain; almigar@ull.edu.es

**Keywords:** aphasia, nursing assessment, standardized nursing terminology, validation studies, surveys and questionnaires

## Abstract

This study presents the development and content validation of an instrument assessing the ability to communicate among individuals with aphasia. The study consists of three stages: (i) Selection and definition of the component dimensions and areas, construction of items assessing these dimensions, administration instructions, and qualitative criteria for assigning diagnoses; (ii) Face validity and content validity; (iii) Pilot test. The tentative questionnaire was designed using two defining characteristics of the NANDA-I (“Impaired verbal communication” and “Readiness for enhanced communication”) and the NOC outcome indicators “Communication”, “Communication: Expressive”, “Communication: Receptive”, and “Information Processing”. The areas and items reached initial content validity index (CVI) and representativeness index (RI) values of 0.87 and above. Those that did not reach the expected values were modified after expert review. The resulting questionnaire was pilot-tested for feasibility and administration times. An instrument containing five dimensions, fourteen areas, and 43 items was obtained and administered in 15 (12–31) minutes. A panel of experts evaluated the final questionnaire (CEECCA), awarding its areas and items CVI and RI values of 0.90 and above. In the absence of further psychometric studies, the questionnaire appears to be useful for assessing ability to communicate in individuals with aphasia.

## 1. Introduction

Aphasia is commonly defined as a loss or impairment of the complex process of interpreting language symbols caused by acquired brain damage [1]. Current definitions consider aphasia to be a multimodal cognitive disorder affecting not only listening comprehension, spoken language, and reading and writing, but also other cognitive processes dependent on the activity of the left cerebral hemisphere, such as auditory-verbal short-term memory and attention, which are indispensable for the proper functioning of language [2]. Cerebrovascular accidents (CVAs) are the most common cause of aphasia [3]. Other aetiological factors include traumatic brain injury (TBI), brain tumours, degenerative diseases, and nervous system infections (NSIs) [4]. Multicentre studies estimating the incidence of post-stroke aphasia carried out in countries such as Switzerland, Italy, Finland, Denmark, and Canada show an incidence of aphasia ranging from 18 to 38% [5,6,7,8]. Aphasia is a condition that generates multiple problems and care needs. A number of studies have shown that aphasia-related impairment or loss of the ability to use language has a considerable impact on quality of life, leading to difficulties in performing activities of daily living [9,10,11] while negatively affecting the individual perception of sufferers and, as a result, their physical, mental, emotional, family, and social functioning. Numerous authors argue that poor communication strategies in healthcare settings increase the impact of aphasia on the loss of personal identity [9,10,11,12,13], resulting in greater isolation and difficulties in accessing social support networks. There is evidence that nurses interacting with individuals with aphasia tend to control and lead the conversation by focusing on nursing tasks and only occasionally allow patients to start a conversation on a topic suited to their own interests and needs [14]. This asymmetric communicative relationship entails a loss of autonomy that makes it difficult for patients to participate in their own care, generating feelings of frustration, fear, and despondency [15,16]. The complexity of aphasia requires multidisciplinary and interdisciplinary interventions in which nurses play a fundamental role as care providers [17]. The quality of nursing care largely depends on proper assessment of patients’ needs. As the first stage in the nursing process, assessment is key to ensuring an accurate diagnosis and a care plan tailored to individual needs. Nursing care is provided in a wide variety of contexts. This makes it necessary for assessment tools to be tailor-made and validated for specific practice settings through tests conducted with the individuals receiving care [18].

Aphasia assessment has generally been the domain of disciplines such as neurology, psychology, linguistics, and speech therapy, and has been addressed from different models and approaches. Most aphasia assessment tests designed within these disciplines require specific expertise in these areas of knowledge. They are also often lengthy tests, constructed from numerous time-consuming subtests [19]. This makes these instruments impractical for use by nurses in their daily work.

Standardized nursing languages (SNLs) have been incorporated into the nursing process as instruments for standardising nursing practice. SNLs provide a useful basis for the development and construction of nursing assessment tools suited to different contexts of nursing practice [20,21]. The most widely used classifications of SNLs in Spain and abroad are: the NANDA-I classification of nursing diagnoses [22], the NOC classification of nursing outcomes [23], and the NIC classification of nursing interventions [24]. All three classifications have a far higher number of validation studies in their favor than other SNLs [25].

The NANDA-I classification of nursing diagnoses provides a way of classifying and categorising areas of nursing professional responsibility [22]. A nursing diagnosis is defined as a clinical judgement of a human response to a health condition or life process (or vulnerability to that response) of an individual, family, group, or community. Nursing diagnoses have a number of key components: a label reflecting the diagnostic focus and nursing judgement; a clear definition of the diagnosis; the defining characteristics of the diagnosis (as observable evidence or inferences grouped as manifestations of the diagnosis); related factors (aetiology, circumstances, events, and/or influences linked to the diagnosis); and risk factors (circumstances increasing the vulnerability of an individual, family, group, or community to suffer an unhealthy event, only present in risk diagnoses) [22,25]. The NANDA-I classification best meets the requirements for nursing diagnosis, as confirmed by several authors [26]. It has been translated into multiple languages and used by thousands of professionals to reflect the care issues underpinning their care work. In recent decades, it has been included in electronic health records as a language that meets the necessary quality standards for documenting nursing diagnoses in various institutions, health services, and organizations such as the Spanish Ministry of Health (Royal Decree 1093/2010 of 3 September) [18].

The NOC classification is intended to identify the outcome criteria and the degree to which these outcomes are achieved for each individual, family, and/or community, and can be used to assess nursing care throughout the entire nursing process [20]. The NOC classification has a taxonomical structure consisting of three levels: Domains, Classes, and Outcomes. It includes indicators for each of the outcomes and measurement scales. Nurse-sensitive patient outcomes are defined as the state, behavior, or perception of a person, family, or community that is measured along a continuum in response to one or more nursing interventions. Each outcome has a set of associated indicators that are used to determine the status of the patient in relation to the outcome [27].

The research hypothesis for this study is that the NANDA-I and NOC standardized nursing languages are useful for the construction of a nursing questionnaire assessing the main areas of language in individuals with aphasia.

The study objectives are as follows: to construct a questionnaire whose dimensions are based on defining characteristics (DCs) of the NANDA-I diagnoses and NOC outcome indicators relating to nurse-patient communication; to compile a bank of items to create the questionnaire; to conduct content validity and representativeness tests; and to assess the feasibility and applicability of the questionnaire obtained.

This instrument is intended to be useful for nurses to assess the main areas of language in individuals with aphasia in a simple manner. This could support effective nurse-patient interaction and care planning linked to activities of daily living and tailored to the ability of patients to communicate.

## 2. Materials and Methods

### 2.1. Design

This is a questionnaire construction and validation study for nurses to assess the ability to communicate among individuals with aphasia. The instrument is designed for use in primary care (PC) and specialized care (SC). The study began in May 2016 and ended in December 2019.

In order to assess face and content validity, a panel of 11 experts was selected by the research team on the basis of their training and experience in relation to the condition in question. Individuals who met at least three of the following requirements within the previous five years qualified as experts:(a)Experience in caring for individuals with aphasia.(b)Accredited training in the care of individuals with aphasia.(c)Teaching or research experience related to the care of individuals with aphasia.(d)Clinical, teaching, or research experience in the care of individuals with acquired brain damage.(e)Accredited training in effective communication techniques.(f)Clinical, teaching, or research experience in the use of SNLs.(g)Participating or having participated in working groups or expert groups in nursing methodology.

The pilot test consisted of an initial administration of the instrument to patients with aphasia. Eight patients were selected using a convenience sampling method in order to cover a wide range of degrees of severity, disorder types, and age groups. They were recruited from specialized rehabilitation centres, hospital neurology departments, and PC facilities. The inclusion criteria were: being at least 18 years old, having been diagnosed with aphasia caused by acquired brain damage, having Spanish as their mother tongue, and agreeing to participate in the study. The exclusion criteria were: having a low level of consciousness, a history of neurological or neurodegenerative disease prior to the brain damage, a psychiatric/psychological history with a communication disorder prior to the brain damage, literacy or cognitive problems preventing them from taking the test, severe visual or hearing impairment, behavioral problems hindering participation, and/or a history of substance abuse.

### 2.2. Phases of Development

#### 2.2.1. Phase 1. Generation of Dimensions, Areas, and Items. A Two-Part Analysis Was Carried out to Select the Dimensions for the Initial Questionnaire

Firstly, a literature search was conducted to identify valid, reliable instruments assessing the main dimensions of language in patients with aphasia. Their design had to be suitable for application by nurses in both hospital and community settings and at any stage of the disease. The instruments identified were reviewed to compare and classify the dimensions of language they assessed.

Secondly, the NANDA-I nursing diagnoses and NOC outcome criteria relating to communication aspects available at the time of the research were analyzed.

The NANDA-I classification includes two communication-related diagnoses [25]:−“Impaired verbal communication” (00051): Decreased, delayed, or absent ability to receive, process, transmit, and/or use a system of symbols;−“Readiness for enhanced communication” (00157): A pattern of exchanging information and ideas with others, which can be strengthened.

The NOC taxonomy includes four communication-related outcomes [27]:−“Communication” (0902): Reception, interpretation, and expression of spoken, written, and non-verbal messages;−“Communication: Expressive” (0903): Expression of meaningful verbal and/or non-verbal messages;−“Communication: Receptive” (0904): Reception and interpretation of verbal and/or non-verbal messages;−“Information Processing” (0907): Ability to acquire, organise, and use information.

Once the DCs of the diagnoses and outcome indicators were selected, they were grouped into language dimensions and then compared with the language dimensions included in the instruments identified in the literature search. The language dimensions included in the preliminary version of the questionnaire were the result of discussion and analysis of the dimensions obtained from the two types of sources.

Thirdly, in order to select the areas making up each dimension, the research team used the Boston Diagnostic Aphasia Examination [28] as a reference, as well as the areas included in the instruments from the literature search. The areas that best represented each dimension according to the purposes of the instrument in the opinion of the research team were selected for the preliminary version of the questionnaire. Care was taken to ensure that the questionnaire was not too long for use in healthcare practice. Items were then designed to assess each area of the questionnaire. A number of them were based on pictograms created by Sergio Palao for the Centro Aragonés para la Comunicación Aumentativa y Alternativa (Aragonese Centre for Alternative and Augmentative Communication), licensed under Creative Commons BY-NC-SA and belonging to the Regional Government of Aragón. This pictographic system was used because of its high degree of simplicity, uniformity, and iconicity. It is also the most widely used set of symbols in Spain and neighbouring countries for alternative and augmentative communication (AAC) [29]. The illustrated concepts were selected from a battery of drawings standardized for use in the Spanish population [30] on the basis of their familiarity and widespread consensus as to their naming. Figure 1 shows an example of the pictograms used by the Aragonese Centre for Alternative and Augmentative Communication to assess the area of verbal naming of objects [31]. Qualitative criteria were also designed to assign dysfunctionality outcomes according to combinations of responses obtained and, in this case, diagnostic labels of communicative impairment. A color code representing different response options was used: red and orange = dysfunctional outcome; yellow and green = functional outcome.

#### 2.2.2. Face and Content Validity

For this phase, as recommended in the literature [32], a panel of 11 experts was selected according to the aforementioned criteria. The variables studied in relation to the experts were: sex, age, and professional field. Each expert evaluated the language dimensions, the areas representing these dimensions, and the relevance and representativeness of the items. Two surveys were prepared and sent consecutively to each expert, 15 days apart. The first survey assessed the relevance (degree to which each item in the questionnaire measures the domain it belongs to); and representativeness (degree of adequacy with which the content of the element represents all the aspects of the domain it belongs to) of the language areas in relation to the overall dimensions of the instrument. The second survey evaluated these same aspects with respect to the items measuring each of the areas. A Likert scale with four response options was used for these evaluations: Not at all relevant/representative; Barely relevant/representative; Quite relevant/representative; Completely relevant/representative. Each response was assigned an ascending score from “Not at all relevant/representative” (0 points) to “Completely relevant/representative” (3 points).

Following the expert test described by Polit et al. [33], once their responses had been received, the content validity index (CVI) for each area and item was calculated by dividing the number of expert evaluations awarded 2 or 3 points by the total number of evaluations for each area and item. Similarly, the representativeness index (RI) was calculated for each area in relation to its dimension and for each item with respect to the area it measured. The overall CVI and RI values for the preliminary version of the questionnaire were calculated using the arithmetic mean for the indices of all the areas and items. Good levels of relevance and representativeness were set at values equal to or greater than 0.78 [33] with a minimum required value of 0.62 for tests with at least ten experts [34]. The areas and items failing to achieve this score were reviewed and modified, taking into account the reasons given by each expert. The CVI and RI were calculated using the following formulae:

Content validity index (CVI) for each area and for each item of the instrument.
$$\mathrm{CVI}=\frac{\mathrm{Number}\;\mathrm{of}\;\mathrm{experts}\;\mathrm{agreeing}\;\mathrm{on}\;\mathrm{the}\;\mathrm{relevance}\;\mathrm{value}\;\mathrm{for}\;\mathrm{each}\;\mathrm{area}/\mathrm{item}\;\left(\mathrm{values}\;\mathrm{between}\;2\;\mathrm{and}\;3\right)\;}{\mathrm{Total}\;\mathrm{number}\;\mathrm{of}\;\mathrm{experts}}$$

Representativeness index (RI) for each area and for each item of the instrument.
$$\mathrm{RI}=\frac{\mathrm{Number}\;\mathrm{of}\;\mathrm{experts}\;\mathrm{agreeing}\;\mathrm{on}\;\mathrm{the}\;\mathrm{representativeness}\;\mathrm{value}\;\mathrm{for}\;\mathrm{each}\;\mathrm{areaitem}\;\left(\mathrm{values}\;\mathrm{between}\;2\;\mathrm{and}\;3\right)\;}{\mathrm{Total}\;\mathrm{number}\;\mathrm{of}\;\mathrm{experts}}$$

Individual content validity index for each expert (CVI-e).
$$\mathrm{CVI}-e=\frac{\mathrm{Number}\;\mathrm{of}\;\mathrm{areas}/\mathrm{items}\;\mathrm{scored}\;\mathrm{between}\;2\;\mathrm{and}\;3\;\mathrm{by}\;\mathrm{an}\;\mathrm{expert}\;}{\mathrm{Total}\;\mathrm{number}\;\mathrm{of}\;\mathrm{areas}/\mathrm{items}}$$

Individual representativeness index for each expert (IR-e)
$$\mathrm{IR}-e=\frac{\mathrm{Number}\;\mathrm{of}\;\mathrm{areas}/\mathrm{items}\;\mathrm{scored}\;\mathrm{between}\;2\;\mathrm{and}\;3\;\mathrm{by}\;\mathrm{an}\;\mathrm{expert}\;}{\mathrm{Total}\;\mathrm{number}\;\mathrm{of}\;\mathrm{areas}/\mathrm{items}}$$

Overall CVI and RI for the preliminary version of the questionnaire:$$\mathrm{Overall}\;\mathrm{CVIRI}=\frac{\mathrm{Sum}\;\mathrm{of}\;\mathrm{CVI}\;\mathrm{and}\;\mathrm{IR}\;\mathrm{from}\;\mathrm{each}\;\mathrm{expert}}{\mathrm{Total}\;\mathrm{number}\;\mathrm{of}\;\mathrm{experts}}$$

#### 2.2.3. Pilot Test

The pilot test assessed questionnaire understandability, administration times, and feasibility. It was carried out in two sub-phases. Once the results of the first administration were analyzed, the necessary changes and adjustments were made to each area and item, generating a modified questionnaire that underwent further expert evaluation to recalculate the CVI and RI values and conduct a second pilot test. This process allowed the instrument to be refined, resulting in an improved evaluation tool. The scope and type of participant recruitment, as well as the inclusion/exclusion criteria are described above.

Informed consent was obtained from the participants prior to the administration of the questionnaire in three ways: informed consent in writing, verbal informed consent in front of a witness, and informed consent from the legal guardian/representative. This ensured compliance with the Spanish Basic Law 41/2002 of 14 November, regulating patient autonomy, rights, and obligations regarding clinical information and documentation, as well as Regulation (EU) 2016/679 of the European Parliament and of the Council of 27 April 2016 on the protection of natural persons with regard to the processing of personal data.

### 2.3. Variables

Data of participants were anonymized using alphanumeric coding so that all clinical data were separated from personal identification data. This process ensures the confidentiality of the data in compliance with the Spanish Organic Law 3/2018 of 5 December, on Personal Data Protection and Guarantee of Digital Rights. In addition to the questionnaire responses, the following sociodemographic and clinical variables were collected: sex, age, place of assessment, level of education, previous reading and writing expression, aetiology of aphasia, type of aphasia, and degree of severity of aphasia according to the Boston examination. This test rates the ability to communicate verbally from 0 (highest severity, with no speech or auditory comprehension) to 5 (lowest severity, with minimal observable speech impairment or subjective difficulties not evident to the assessor).

### 2.4. Data Analysis

The results of the preliminary revised version of the questionnaire after the two pilot tests, as well as the data on sociodemographic and clinical variables, were stored in an SPSS v.25.0 database for processing. Nominal variables were summarized as absolute and relative frequencies of their categories, and quantitative variables were expressed as medians (minimum-maximum) as they are not normally distributed due to the sample size.

## 3. Results

### 3.1. Phase 1

The following Medical Subject Headings (MeSH) descriptors were selected for the literature search: Nursing; Aphasia; Aphasia Test; Neuropsychological Test; Acquired Language Disorder; Assessment of Health Care Need; Nursing Care. The databases consulted were: PubMed (MedLine); Scopus; Web of Science, Biblioteca Virtual en Salud España (BVS); IBECS; CUIDEN; EMBASE; Dialnet; SciELO España; Biblioteca Cochrane Plus; CINAHL; and Science Direct. We searched for publications in English and Spanish, in the title and abstract fields, with no date limit, using keywords and MeSH terms. The search strategies were: (“neuropsychological tests” OR “neuropsychological” AND “tests”) AND (“aphasia”); “aphasia test”; “aphasia assessment”; “aphasia battery”; “aphasia examination”; “aphasia” AND “nursing”; “aphasia” AND “assessment” AND “nursing”; “aphasia” AND (“physical examination” OR “physical” AND “examination”) AND “nursing”; “neuropsychological test” AND “nursing”; “aphasia” AND “assessment” AND (“delivery of health care”) OR (“delivery” AND “health” AND “care”) OR (“delivery of health care”) OR (“health”AND “care”) OR (“health care” AND “need”).

Eight studies were selected for inclusion in the dimension elicitation phase of the preliminary questionnaire. Of these, seven studies reported on the construction and validation of aphasia screening instruments [35,36,37,38,39,40,41]. The other study addressed the adaptation and validation of a screening instrument [42].

Among the nursing diagnoses of the NANDA-I 2015–2017 classification concerning communicative aspects, “Impaired verbal communication” (00051) and “Readiness for enhanced communication” (00157) were selected as being of interest to this research as their wording could be related to individuals with aphasia. The DCs of these diagnoses were classified according to language dimensions. The research team reviewed the latest available editions of the NANDA-I and agreed to use the 2012–2014 edition to classify the DCs of the diagnosis “Readiness for enhanced communication” (00157) as it was more fully developed for the purposes of this study.

The same procedure was carried out with the four NOC outcome criteria identified in relation to language and communication: “Communication” (0902), “Communication: Expressive” (0903), “Communication: Receptive” (0904), “Information Processing” (0907). The indicators potentially present in individuals with aphasia were selected and classified according to language dimensions. The use of the NOC increased the number of dimensions with respect to those obtained from the NANDA-I diagnoses. Table 1 shows the distribution of the DCs of the diagnoses and the outcome indicators by language dimensions.

The research team then agreed on five dimensions for the preliminary questionnaire: the dimensions that coincided with those obtained from the analysis and review of aphasia assessment instruments in the literature search and those obtained from the analysis and classification of NANDA-I DCs and NOC indicators. Thus, the preliminary version of the questionnaire, which was tested for content validity, comprised the following dimensions: Verbal expression; Written expression; Expression through drawings, illustrations, and icons; Auditory comprehension; and Reading comprehension. In turn, the dimensions were divided into 13 areas, assessed by 39 items: 25 on expression areas and 14 on comprehension areas. Qualitative criteria were devised to assign functional/dysfunctional categories and communicative impairment diagnostic labels according to the results obtained by combining responses.

Table 2 shows the structure of the preliminary questionnaire, arranged in dimensions, areas, and items.

### 3.2. Phase 2

Of the eleven experts, seven were men, with a mean age of 43 years; six were employed in healthcare, five in teaching and research, and two in management. Four worked in PC, five in SC, and two at the university. The experts evaluated all areas and items of the draft version of the questionnaire. Both the overall CVI and RI values for the areas were 0.87. The CVI and RI values for each area and item, as well as the overall values of the questionnaire, are shown in Table 2.

The following modifications were made after the qualitative review by the group of experts:−The area “Expression through drawings, illustrations, and icons” was renamed “Expression through pictograms” to more accurately represent the task at hand.−A new item, scissors, was added to the area “Naming objects verbally”, increasing the number of semantic categories represented.−In the area “Naming actions verbally”, item 12, urinating, was replaced by a new pictogram representing the action of combing one’s hair in order to avoid sex-based differences.

These modifications were qualitatively analyzed by the entire group of experts, who, in a final brainstorming session, reached a consensus to accept the second version of the tentative questionnaire, known as CEECCA, Cuestionario para la Evaluación Enfermera de las Capacidades Comunicativas en la Afasia (Nursing Assessment of Ability to Communicate among Patients with Aphasia Questionnaire).

### 3.3. Phase 3

The first phase of the pilot test involved six participants, 50% female, with a mean age of 66 (45–81) years (Table 3). Five participants were assessed at a comprehensive rehabilitation centre and one at a tertiary hospital. One participant had primary education, two had vocational training, and two had university education. The aetiology of aphasia was ischaemic stroke in three participants, haemorrhagic stroke in two participants, and brain tumour in one participant. The sample included one case of motor aphasia, one case of global aphasia, one case of anomic aphasia, two cases of transcortical sensory aphasia, and one case of transcortical motor aphasia. According to the Boston examination, their degree of severity obtained a median score of 3 (0–4) points. The first administration lasted a mean of 12 (11–29) min.

The research team assessed the results from the first phase of the pilot test of the CEECCA and made modifications to its content and wording. As a result, the area “Auditory comprehension of sentences” was included in the “Auditory comprehension” dimension, and response time limits were established as a criterion for dysfunctionality.

The resulting third version of the questionnaire was re-evaluated by the panel of experts, with its areas obtaining a CVI of 0.96 and an RI of 0.90, and items obtaining overall CVI and RI values of 0.90. This was followed by the second phase of the pilot test, with the questionnaire being administered to six participants, four from the previous phase and two new ones. One was recruited and assessed at a specialized rehabilitation centre with transcortical motor aphasia following central nervous system infection, and another at a PC facility with anomic motor aphasia following ischaemic stroke. In this sample, 50% were women, the mean age was 69 (52–82) years, and the degree of severity was 2 (0–3) points (Table 3). The administration time was 16 (12–31) min.

Table 4 shows the dimensions, areas, and items in the third and final version of the CEECCA questionnaire, as well as the criteria for assigning diagnostic labels and the percentages of dysfunctionality by area of participants.

## 4. Discussion

The main outcome of this study is the development of the CEECCA instrument based on the standardized nursing languages NANDA-I and NOC which enables nurses to assess the main dimensions of language in individuals with aphasia in a simple manner. Further studies should test its psychometric properties with a sufficiently large sample of individuals with aphasia. The CEECCA consists of 43 items assessing 14 specific language areas corresponding to five global dimensions likely to be affected in these patients. Language assessment using the CEECCA is intended, firstly, to promote effective interactions between the nurse and the patient with aphasia; secondly, to serve as a valid, reliable tool for making diagnostic judgements, and thirdly, to supplement the assessment of outcome indicators. The use of the NANDA-I and NOC classifications to construct the questionnaire ensured that the selected dimensions were relevant to nursing practice.

The literature search revealed only one validated screening instrument for use by nurses [36]. The instrument is intended to detect aphasia in acute cerebrovascular events and to provide information on overall language status across four categories: no language disorder, mild language disorder, moderate language disorder, and severe language disorder. In turn, each area is assessed in terms of dichotomous variables: normal response or impaired response. In our opinion, this categorization of responses does not offer sufficient information about the degree of functionality of the different language areas, as one area may be affected and yet be functional enough to generate a communicative exchange. On the other hand, the CEECCA aims to serve as a tool for assessing language in patients already diagnosed with aphasia, providing information on functionality in various dimensions. In this sense, responses are classified according to three or four levels of competence, which are in turn categorized as functional or dysfunctional. This approach aims to identify those responses which, while not entirely correct, can be functional in generating a communicative exchange.

The CEECCA questionnaire dispenses with two dimensions included in most of the instruments reviewed for assessing aphasia: repetition and automatic or serial speech. On its own, a repetition test does not provide a non-specialist assessor with relevant information about the ability to maintain a functional communicative exchange [43]. Automatic or serial speech may be preserved in patients with severe spontaneous speech impairment [44] and does not provide information on the ability to attach meaning to the spoken word, either receptively or expressively [28].

The dimension “Expression through pictograms”, obtained from the NOC outcomes, is viewed as particularly relevant. When aphasia is chronic or severe, the patient may benefit from AAC [45], as it improves communication skills and social engagement [46,47]. There are multiple AAC strategies, which are difficult to represent in instruments such as the CEECCA. However, we believe that the items designed to assess this dimension provide valuable information on whether or not the patient can benefit from such a system.

The overall CVI and RI values for both areas and items suggest high levels of relevance and representativeness of the instrument [33].

The CEECCA administration times are within the intervals required to complete other tests designed using the NANDA-I and NOC classifications [48,49,50]. After the first sub-phase of the pilot test, the need to limit the response time for each item was considered. As a result, response times were established as a criterion for dysfunctionality. Most individuals with aphasia have increased response times. Instruments such as the Boston examination [28] and the Western aphasia battery [51] limit response times for a number of verbal tasks. During the first sub-phase of our pilot test, it was noted that administration of the questionnaire proved excessively lengthy as no time limits had been included. The evaluators also struggled to adhere to the established times in the pilot test phase. Once the time limit has been exceeded without producing a response and the response has therefore been classified as dysfunctional, the assessor should encourage the patient to produce or generate some type of response by providing some kind of assistance. If the examiner simply moves from one item to the next without prompting the patient to produce a response, this may lead to feelings of helplessness or discouragement, increasing the total test administration time.

This study has a number of limitations. Firstly, the number of experts was sufficient but could have been expanded to include professionals from other backgrounds besides nurses to provide a multidisciplinary vision of this validation phase. Nevertheless, our aim was to obtain a nursing assessment tool. Secondly, the limited sample size (*N* = 8) with which the pilot test of this questionnaire was conducted must be taken into consideration. It goes without saying that the rest of the psychometric tests of validity, consistency, and reliability of the resulting questionnaire will have to be conducted with a larger representative sample. In our opinion, the circumstances that limited the recruitment of a larger sample size for the pilot test include the temporal and operational limitations of the study and the difficulty in recruiting participants with this clinical and neuropsychological profile.

When looking at the samples with which a number of screening instruments for diagnosing aphasia have been validated, we observe small sample sizes, most of them below 50 subjects [35,36,39,42]. Most of them fail to mention the number of subjects with whom the preliminary questionnaires were pilot-tested and do not discuss why such limited samples of subjects were used to conduct their psychometric tests. Several publications refer to the challenge of obtaining informed consent from individuals with language and communication disorders. This has led to a systematic exclusion of individuals with aphasia from the samples of a large number of studies, especially from studies assessing different conditions related to CVAs and acquired brain injury [52].

Thirdly, the concepts comprising the items corresponding to the areas “Naming objects verbally”, “Naming objects in writing”, “Auditory comprehension of words”, and “Reading comprehension of words” were selected on the basis of their high familiarity, excluding concepts of medium and low familiarity. This could influence performance in these areas, with concepts being evoked with less difficulty [43]. We decided to assess basic, less demanding processes in order to identify opportunities for interacting with patients with severe language impairment. This entails a potential loss of sensitivity to patients with mild aphasic disorders, which will have to be addressed in future revisions of the questionnaire. Finally, we were aware that it may not be possible to obtain specific NANDA-I diagnoses for the problems detected using the CEECA. However, the construction of the instrument itself offers up to ten new diagnostic proposals (Table 4) that could be included in future NANDA-I classifications once the psychometric properties of the CEECCA have been tested and these labels have been validated in patients with these characteristics.

## 5. Conclusions

This study presents an instrument, the CEECCA, constructed using DCs from NANDA-I diagnoses and NOC outcome indicators, which shows adequate content validity and representativeness. The instrument has proved useful for nurses seeking to assess ability to communicate among individuals with aphasia in PC and SC settings. This study illustrates the utility of the NANDA-I and NOC SNLs in constructing instruments to improve the accuracy of nursing diagnoses and supplement the measurement of outcome indicators based on a solid conceptual framework, targeting a specific setting, and validated through tests carried out with patients who will receive such care. Given the relevance of the communicative interaction between nurses and patients as the backbone of the care process itself, the CEECCA can help nurses to adapt their modes of communication to the actual abilities of each patient, with the possibility of using AAC to increase the quality of their interactions. Due to the relationship between severity of language impairment and other factors, the CEECCA results could serve as predictors of major dysfunctions or as facilitators for nursing interventions to address social isolation, anxiety, hopelessness, and risk of loneliness, among other human responses.

## Figures and Tables

**Figure 1 healthcare-09-01459-f001:**
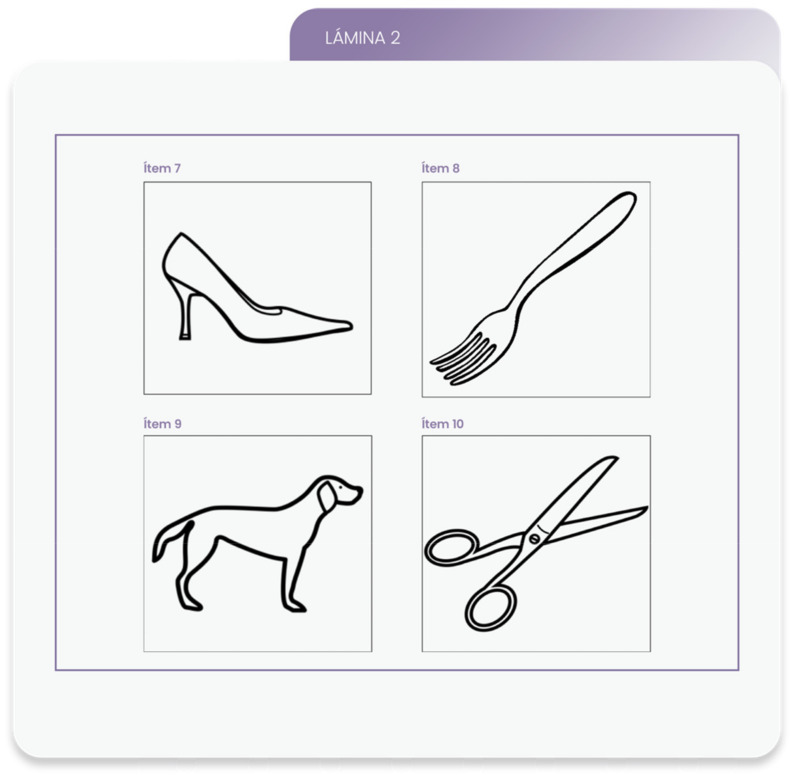
Pictograms used by the Aragonese Centre for Alternative and Augmentative Communication to assess the area of verbal naming of objects.

**Table 1 healthcare-09-01459-t001:** Defining characteristics of the NANDA-I diagnoses and the NOC outcome indicators distributed by language dimensions.

NANDA-I Diagnoses		NOC Outcome Criteria				Resulting Language Dimensions
Impaired Verbal Communication (00051)	Readiness for Enhanced Communication (00157)	Communication (0902)	Communication: Expressive (0903)	Communication: Receptive (0904)	Information Processing (0907)	
Defining Characteristics		NOC Outcome Indicators				
− Difficulty maintaining communication. − Difficulty expressing thoughts verbally. − Difficulty forming words. − Difficulty forming sentences. − Difficulty speaking. − Difficulty verbalising. − Inability to speak. − Slurred speech. − Inappropriate verbalization.	− Able to speak a language. − Forms words. − Forms sentences. − Expresses thoughts. − Expresses feelings. − Expresses desire to enhance communication.	− Use of spoken language. − Directs messages to correct recipient. − Exchanges messages accurately with others.	− Use of spoken language: vocal. − Clarity of speech.		− Verbalises a coherent message. − Exhibits organized thought processes. − Exhibits logical thought processes. − Explains similarity between two items. − Explains dissimilarity between two items.	Verbal expression
− Difficulty comprehending communication. − Difficulty maintaining communication.		− Acknowledgement of messages received. − Accurate interpretation of messages received. − Exchanges messages accurately with others.		− Interpretation of spoken language. − Acknowledgement of messages received.	− Comprehends a sentence. − Comprehends a paragraph. − Comprehends a story. − Comprehends universal symbols. − Exhibits organized thought processes. − Exhibits logical thought processes.	Auditory comprehension
					− Identifies common objects.	Naming skills
	− Able to write a language.	− Use of written language.	− Use of written language.			Written expression
				− Interpretation of written language.		Reading comprehension
− Difficulty in use of body expressions. − Difficulty in use of facial expressions. − Inability to use body expressions. − Inability to use facial expressions.	− Uses non-verbal cues appropriately.	− Use of non-verbal language.	− Use of non-verbal language.			Facial and body expression
	− Interprets non-verbal cues appropriately.			− Interpretation of non-verbal language. − Interpretation of sign language.		Comprehension of facial and body language
		− Use of pictures or drawings.	− Use of pictures and drawings. − Use of sign language.			Expression through symbols, illustrations, and icons
				− Interpretation of pictures and drawings.		Language comprehension through symbols, illustrations, and icons

**Table 2 healthcare-09-01459-t002:** Dimensions, areas, and items of the CEECCA questionnaire in its preliminary and final versions. CVI and RI values for each area and item.

Dimensions in the Preliminary Questionnaire	Areas and Items in the Preliminary Questionnaire	CVI/RI Values for Each Area and Item in the Preliminary Questionnaire		Areas and Items in the Final Questionnaire	CVI/RI Values for Each Area and Item in the Final Questionnaire	
Verbal expression	Conversational speech:	1.00	1.00	Conversational speech:	1.00	1.00
	− Item 1. Open-ended question: “How are you feeling today?” − Item 2. “What’s your full name?” − Item 3. “What’s your address? Where do you live?” − Item 4. Yes/no answer. “Did you get a good night’s rest?” − Item 5. Open-ended question. “Please tell me what your work consists of/consisted of.”	0.91 1.00 0.82 0.73 0.82	0.91 0.91 0.82 0.82 0.82	− Item 1. Open-ended question: “How are you feeling today?” − Item 2. “What’s your full name?” − Item 3. “What’s your address? Where do you live?” − Item 4. Yes/no answer. “Did you get a good night’s rest?” − Item 5. Open-ended question. “Please tell me what your work consists of/consisted of.”	0.91 1.00 0.91 0.82 0.91	0.91 1.00 0.91 0.82 0.91
	Descriptive speech:	0.82	0.82	Descriptive speech:	1.00	0.91
	− Item 6. Describing an illustrated scene.	0.82	0.73	− Item 6. Describing an illustrated scene.	0.91	0.91
	Naming objects verbally (visual confrontation naming):	1.00	1.00	Naming objects verbally (visual confrontation naming):	1.00	1.00
	− Item 7. “Shoe” − Item 8. “Fork” − Item 9. “Dog”	0.91 0.91 0.91	0.91 0.91 0.91	− Item 7. “Shoe” − Item 8. “Fork” − Item 9. “Dog” − Item 10. “Scissors”	0.91 0.91 0.91 0.91	0.91 0.91 0.91 0.91
	Naming actions verbally (visual confrontation naming):	0.91	0.91	Naming actions verbally (visual confrontation naming):	1.00	1.00
	− Item 10. “Eating” − Item 11. “Sleeping” − Item 12. “Urinating”	0.91 0.91 0.82	0.91 0.91 0.73	− Item 11. “Eating” − Item 12. “Sleeping” − Item 13. “Combing”	0.91 0.91 0.91	0.82 0.91 0.91
Written expression	Writing the full name:	0.73	0.73	Writing the full name:	0.82	0.82
	− Item 13. Writing full name and surname(s)	0.73	0.91	− Item 14. Writing full name and surname(s)	0.91	0.91
	Naming objects in writing (written confrontation naming):	0.73	0.82	Naming objects in writing (written confrontation naming):	0.91	0.91
	− Item 14. “Shirt” − Item 15. “Chair” − Item 16. “Dog”	0.91 0.91 0.91	0.91 0.91 0.91	− Item 15. “Shirt” − Item 16. “Chair” − Item 17. “Dog”	0.91 0.91 0.91	0.91 0.91 0.91
	Naming actions in writing (written confrontation naming):	0.91	0.91	Naming actions in writing (written confrontation naming):	0.91	0.91
	− Item 17. “Drinking” − Item 18. “Sleeping” − Item 19: “Running”	0.91 0.91 0.91	0.91 0.91 0.91	− Item 18. “Drinking” − Item 19. “Sleeping” − Item 20: “Running”	0.91 0.91 0.91	0.91 0.91 0.91
Expression through symbols, illustrations, and icons Expression through pictograms	Expressing actions through pictograms (pointing to the correct picture):	0.91	0.91	Expressing actions through pictograms (pointing to the correct picture):	1.00	1.00
	− Item 20. “Eating” − Item 21. “Taking a shower” − Item 22. “Reading”	0.91 0.91 0.91	0.91 0.91 0.91	− Item 21. “Eating” − Item 22. “Taking a shower” − Item 23. “Leer”	0.91 0.91 0.91	0.91 0.91 0.91
	Expressing emotions through pictograms (pointing to the correct picture):	0.91	0.82	Expressing emotions through pictograms (pointing to the correct picture):	0.91	0.91
	− Item 23. “Happy” − Item 24. “Sad” − Item 25. “Angry”	0.91 0.91 0.91	0.91 0.91 0.91	− Item 24. “Happy” − Item 25. “Sad” − Item 26. “Angry”	0.91 0.91 0.91	0.91 0.91 0.91
Auditory comprehension	Auditory comprehension of words. Matching spoken words with pictures (pointing to the correct picture):	1.00	0.91	Auditory comprehension of words. Matching spoken words with pictures (pointing to the correct picture):	1.00	0.82
	− Item 26. “Comb” − Item 27. “Fork” − Item 28. “Pear” − Item 29. “Hand” − Item 30. “Trousers”	1.00 1.00 0.91 0.91 1.00	0.91 0.91 0.73 0.91 0.91	− Item 27. “Comb” − Item 28. “Fork” − Item 29. “Pear” − Item 30. “Hand” − Item 31. “Trousers”	1.00 1.00 1.00 1.00 1.00	0.91 0.91 0.82 0.82 0.82
				Auditory comprehension of sentences. Selecting the picture corresponding to the sentence expressed verbally (pointing to the correct picture):	1.00	0.82
				− Item 32. Select one of the three sentences to tell the patient: ◦ The bench is surrounded by three lampposts. ◦ The lamppost is surrounded by three benches. ◦ The bench is between two lampposts. − Item 33. Select one of the three sentences to tell the patient: ◦ The ball is under the table that is between the chair and the cupboard. ◦ The ball is on the chair that is between the table and the cupboard. ◦ The ball is on the table that is between the cupboard and the chair. − Item 34. Select one of the three sentences to tell the patient: ◦ The big car moves behind the bicycle and in front of the small car. ◦ The big car moves behind the small car and in front of the bicycle. ◦ The small car moves behind the bicycle and in front of the big car.	0.91 1.00 1.00	0.91 1.00 1.00
	Auditory comprehension of verbal commands. Execution of verbal commands:	1.00	0.91	Auditory comprehension of verbal commands. Execution of verbal commands:	1.00	0.91
	− Item 31. “Please look at the ceiling” − Item 32. “Please raise your hand, then pick up the pencil” − Item 33. “Please touch your ear, then touch your nose, and close your eyes”	1.00 1.00 1.00	1.00 1.00 1.00	− Item 35. “Please look at the ceiling” − Item 36. “Please raise your hand, then pick up the pencil (or pen)” − Item 37. “Please touch your ear, then touch your nose, and close your eyes”	1.00 1.00 1.00	1.00 1.00 1.00
Reading comprehension	Reading comprehension of words. Matching written words and pictures:	0.91	0.91	Reading comprehension of words. Matching written words and pictures:	1.00	0.82
	− Item 34. “Fork” − Item 35. “Apple” − Item 36. “Door”	0.91 0.91 0.82	0.91 0.91 0.82	− Item 38. “Apple” − Item 39. “Fork” − Item 40. “Door”	0.82 0.91 0.91	0.82 0.91 0.91
	Reading comprehension of sentences. Complete the sentence with a correct word choice:	0.91	0.91	Reading comprehension of sentences. Complete the sentence with a correct word choice:	0.91	0.82
	− Item 37. “A chair has four…” − Item 38. “To make an omelette I need…” − Item 39. “Juan is a musician, he spends many hours singing and playing the…”	0.91 0.91 0.91	0.91 0.91 0.91	− Item 41. “A chair has four…” − Item 42. “To make an omelette I need…” − Item 43. “Juan is a musician, he spends many hours singing and playing the…”	0.91 0.91 0.91	0.91 0.91 0.91
Overall CVI/Overall RI		0.90	0.89	Overall CVI/Overall RI	0.90	0.90

**Table 3 healthcare-09-01459-t003:** Main demographic and clinical characteristics of the participants included in the two phases of the pilot test.

Participant (Code)	Sex	Age (Years)	Level of Education	Place of Assessment	Type of Aphasia	Aphasia Aetiology	Previous Reading Level	Previous Writing Level	Level of Severity (0–5) Severity Scale: the Boston Test
A1	Male	51	Vocational training	CREN Rehabilitation Centre	Motor aphasia	Ischaemic CVA	Good	Good	1
A2	Female	81	University education	CREN Rehabilitation Centre	Transcortical motor aphasia	Ischaemic CVA	Very good	Very good	3
A3	Male	58	Vocational training	CREN Rehabilitation Centre	Transcortical sensory aphasia	Haemorrhagic CVA	Good	Good	2
A4	Female	78	Primary education	CREN Rehabilitation Centre	Global aphasia	Ischaemic CVA	Good	Good	0
A5	Female	44	University education	CREN Rehabilitation Centre	Anomic aphasia	Haemorrhagic CVA	Good	Good	4
A6	Male	74	Primary education	Nuestra Señora de Candelaria University Hospital	Transcortical sensory aphasia	Brain tumour	Average	Average	4
A7	Male	58	University education	CREN Rehabilitation Centre	Motor aphasia	CNS infection	Very good	Very good	2
A8	Female	78	Primary education	Primary care	Anomic motor aphasia	Ischaemic CVA	Average	Average	1

Participants included in the first phase of the pilot test: A1, A2, A3, A4, A5, A6. Participants included in the second phase of the pilot test: A1, A2, A3, A4, A7, A8.

**Table 4 healthcare-09-01459-t004:** Dimensions, areas, and items in the final CEEECA questionnaire. Diagnostic labels and criteria for assigning them. Dysfunctionality percentages after the second phase of the pilot test.

Dimensions	Areas in the Preliminary CEECCA Questionnaire	Items	Diagnostic Label Assignment Criteria	Diagnostic Labels	Area Dysfunctionality (%)
Verbal expression	Conversational speech	Items 1–5	If two or more responses are dysfunctional	Impaired verbal communication: Conversational speech	33.3
	Descriptive speech	Item 6	If the response is dysfunctional	Impaired verbal communication: Descriptive speech	83.3
	Naming objects verbally	Items 7–10	If two or more responses are dysfunctional	Impaired verbal naming: Objects	50.0
	Naming actions verbally	Items 11–13		Impaired verbal naming: Actions	50.0
Written expression	Writing name and surname(s)	Item 14	If the response is dysfunctional	Impaired written expression: Writing name	83.3
	Naming objects in writing	Items 15–17	If two or more responses are dysfunctional	Impaired written naming: Objects	66.7
	Naming actions in writing	Items 18–20		Impaired written naming: Actions	83.3
Expression through pictograms	Expressing actions through pictograms	Items 21–23	If two or more responses are dysfunctional	Impaired expression through pictograms: Actions	16.7
	Expressing emotions through pictograms	Items 24–26		Impaired expression through pictograms: Emotions	16.7
Auditory comprehension	Auditory comprehension of words	Items 27–31	If two or more responses are dysfunctional	Impaired auditory comprehension: Words	83.3
	Auditory comprehension of sentences	Items 32–34	If two or more responses are dysfunctional; a dysfunctional response if item 32 is involved	Impaired auditory comprehension: Sentences	66.7
	Auditory comprehension of commands	Item 35–37	If two or more responses are dysfunctional; a dysfunctional response if item 35 is involved	Impaired auditory comprehension: Verbal commands	50.0
Reading comprehension	Reading comprehension of words	Items 38–40	If two or more responses are dysfunctional	Impaired reading comprehension: Words	33.3
	Reading comprehension of sentences	Items 41–43		Impaired reading comprehension: Sentences	66.7

## Data Availability

The data presented in this study are available upon request from the corresponding author. The data are not publicly available due to privacy/ethical restrictions.

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
