# Peer review of "Development and Content Validation of the CEECCA Questionnaire to Assess Ability to Communicate among Individuals with Aphasia Based on the NANDA-I and NOC"

_healthcare, 2021, doi:10.3390/healthcare9111459_

Round 1
Reviewer 1 Report
Thanks for recommending me as a reviewer. This study presents the development and content validation of an instrument assessing the ability to communicate among individuals with aphasia. If the authors complete the revision, the quality of the study will be further improved.
- The introduction section is well written. If the authors describes more specifically the theoretical background of "NANDA-I diagnoses" and "NOC outcome indicators" in the introduction section, it can help readers understand.
2. line 74-80: I suggest that the author state the research objective in a sentence in the introduction section.
3. line 82-88: I suggest combining "2.1. Design", "2.2. Setting and Participants", and "2.2.1. Sample of Professionals" into one paragraph.
4. line 102: "2.2.2. Sample of Participants " - Authors should be more specific about the characteristics of the subject.
Author Response
Consulte el archivo adjunto

Reviewer 2 Report
Dear Authors,
Please see the attachment.

Author Response
Consulte el archivo adjunto
